# Accelerating dynamics of collective attention

Philipp Lorenz-Spreen [1,2], Bjarke Mørch Mønsted [3], Philipp Hövel [1,4] & Sune Lehmann [3,5]

With news pushed to smart phones in real time and social media reactions spreading across the globe in seconds, the public discussion can appear accelerated and temporally fragmented. In longitudinal datasets across various domains, covering multiple decades, we find increasing gradients and shortened periods in the trajectories of how cultural items receive collective attention. Is this the inevitable conclusion of the way information is disseminated and consumed? Our findings support this hypothesis. Using a simple mathematical model of topics competing for finite collective attention, we are able to explain the empirical data remarkably well. Our modeling suggests that the accelerating ups and downs of popular content are driven by increasing production and consumption of content, resulting in a more rapid exhaustion of limited attention resources. In the interplay with competition for novelty, this causes growing turnover rates and individual topics receiving shorter intervals of collective attention.

[1] Institute of Theoretical Physics, Technische Universität Berlin, Hardenbergstraße 36, 10623 Berlin, Germany. [2] Center for Adaptive Rationality, Max Planck Institute for Human Development, Lentzeallee 94, 14195 Berlin, Germany. [3] Department of Applied Mathematics and Computer Science, Technical University of Denmark, Richard Petersens Plads, 2800 Kgs., Lyngby, Denmark. [4] School of Mathematical Sciences, University College Cork, Western Road, Cork T12 XF62, Ireland. [5] Center for Social Data Science, University of Copenhagen, Øster Farimagsgade 5, 1353 Copenhagen K., Denmark. These authors contributed equally: Philipp Hövel & Sune Lehmann. Correspondence and requests for materials should be addressed to P.H. (email: philipp.hoevel@ucc.ie) or to S.L. (email: sljo@dtu.dk)

It has long been argued quantitatively that technological developments are accelerating across a large number of domains from genomic sequencing[1] to computational power[2] and communication[3]. Described as part of a more general development termed social acceleration, the impact of these changes on the social sphere has more recently been discussed within sociology[4]. In the literature there have been hints of acceleration in different contexts[5–7], but so far, the phenomenon lacks a strong empirical foundation[8]. While still new relative to many elements of contemporary society, big data has been a part of our culture long enough to enable measurements of longitudinal changes of our collective behavior[9]. Here, we propose a simple way to measure the pace of the dynamic allocation of collective attention through the ebbs and flows of popular topics and cultural items. By that we focus on one dimension of social acceleration, the increasing rates of change within collective attention[10].

As an illustrative example, we begin by studying Twitter, which captures recent years (2013–2016) with high temporal resolution, then expand our analysis to a number of other systems. Our data sources cover multiple decades, as in Google Books (100 years), movie ticket sales (40 years), or citations of scientific publications (25 years). For the recent past, online media enable analysis with higher temporal granularity through Google Trends (2010–2018), Reddit (2010–2015), and Wikipedia (2012–2017) (see also Supplementary Table 1).

We use these dynamical measurements of categorized content as proxies for the amount of collective attention a topic receives. Thus, we use the term collective attention in a sense that is closely related to the computational social science notion[11–13], that measures various forms of population-level content consumption patterns[14–16]. This is distinct from the definition of attention used in cognitive psychology[17,18], which describes the selective allocation of cognitive resource on an individual level.

Across the different domains under investigation, we find clear empirical evidence of ever steeper gradients and shorter intervals of collective attention given to each cultural item. By studying a variety of social domains, we emphasize the general nature of the observed development across both, domains and over time.

To probe the mechanisms underlying these dynamics, we model topics competing for attention[19,20], driven by imitation[21,22] and saturation[12,13,15,23,24], as a function of communication rates. Combining these factors in a minimal mathematical model based on modified Lotka–Volterra[25,26] dynamics reproduces the empirical findings surprisingly well. Our analysis suggests increasing rates of content production and consumption as the most important driving force for the accelerating dynamics of collective attention. The resulting picture is an attention economy[11], where the increasing abundance of information combined with the cognitive limitations and time constraints of users, leads to a redistribution of the available resources across time towards more rapid changes and higher frequencies.

## Results

**Hashtag peaks become increasingly steep and frequent**. In order to measure the pace of collective attention, we start by defining observables to assess its dynamics. As a proxy for the popularity of a topic in Twitter, we use the time series of occurrence counts $L_i(t)$ of hashtag $i$ (Fig. 1a). The variable $L$ refers to the term like as a measure for online-popularity and counts the number of usages of one hashtag $i$ per time $t$ (by any user). Here hashtags serve as a rough classification of content into different topics. We use a time resolution of 24 h in order to avoid noise from diurnal rhythms and the influence of different timezones.

We consider the global top 50 used hashtags, sampled every recorded hour from a corpus of ~43 billion tweets[27] (our results are robust to the specific size of the top group, see Supplementary Fig. 1). By considering top groups of fixed size we ensure the same number of data points in each aggregation window, avoiding bias from increasing counts of simultaneous events. The choice of top groups also maximizes the size of the contributing population and minimizes the influence of locally heterogeneous user groups. Thus, we focus on globally visible topics, discussed around the world.

An early indication of systematic changes is growing transit rates of different hashtags through the top group. The numbers of hashtags passing through the top 50 (Supplementary Table 2) show that in 2013 a hashtag stayed within the top 50 for 17.5 hours on average, a number which gradually decreases to 11.9 hours in 2016 (details in the Methods section and Supplementary Table 2).

For more detailed investigations about long-term developments at a macroscopic level, we focus on statistical properties of the trajectories $L_i(t)$. The ensemble average trajectory of all hashtags $\langle L(t - t_{\text{peak}}) \rangle$ (see Methods section for details) for the years 2013 to 2016, around local maxima $L_i(t_{\text{peak}})$, reveals a key property of the change in the dynamics of collective attention (Fig. 1b). The average maximum popularity $\langle L(t_{\text{peak}}) \rangle$ on one day $t_{\text{peak}}$ stays relatively constant, while the average gradients $\langle \Delta L \rangle$ in positive and negative direction become steeper over the years. The time required to achieve peak popularity decreases and the descent occurs symmetrically faster, shortening the period of each topic's popularity.

To visualize the temporal density of the local maxima $L_i(t_{\text{peak}})$ we define bursts as extreme slopes toward a local maximum, directly followed by a fast decline (see Methods section for details). Figure 1c illustrates the size and the timing of burst occurrences, showing an increasing density of events from bottom to top.

**Distributions of gains and losses are growing broader**. Motivated by these qualitative observations, we define measures for velocities. The main observables for our investigation are the discrete empirical growth rates[16,28]. We divide them into relative gains $[\Delta L_i^{(g)}/L_i](t) = (L_i(t) - L_i(t-1))/L_i(t-1) > 0$ and relative losses $[\Delta L_i^{(l)}/L_i](t) = (L_i(t) - L_i(t+1))/L_i(t+1) > 0$ of each hashtag $i$ and for all available time points $t$. Intentionally independent of the system size, relative velocity measures allow comparison across long time intervals and across different domains. An alternative measure is the logarithmic change log $(L_i(t)/L_i(t-1))$, which weighs large events less strongly, but provides similar results.

We analyze the two distributions of all losses and gains at all times in Fig. 1d, e, respectively. A good fit for both distributions is the log-normal distribution $P(x) = 1/(x\sigma\sqrt{2\pi}) \exp\left[-(\ln(x) - \mu)^2/(2\sigma^2)\right]$ (see Methods section and Supplementary Fig. 2 for alternative probability distribution functions). We find growing values of the fitted parameters $\sigma$ and $\mu$ (values and testing results in Supplementary Table 3). The changes in $\sigma$ and $\mu$ clearly quantify a systematic shift as well as the increasing skewness of the distributions, supporting the visual observations in Fig. 1b–e. Based on the formal definition of acceleration, we interpret changes of the logarithmic derivative $\Delta L/L$ to correspond to a nonzero second derivative of $L_i(t)$, i.e., an acceleration of collective attention dynamics. Since we observed stable average peak heights in Fig. 1b, we show the full distribution of maxima $P(L_i(t_{\text{peak}}))$ in the inset of Fig. 1e. The distribution is very broad (power-law like) and importantly, stays stable over time, corroborating the constant average value from Fig. 1b.

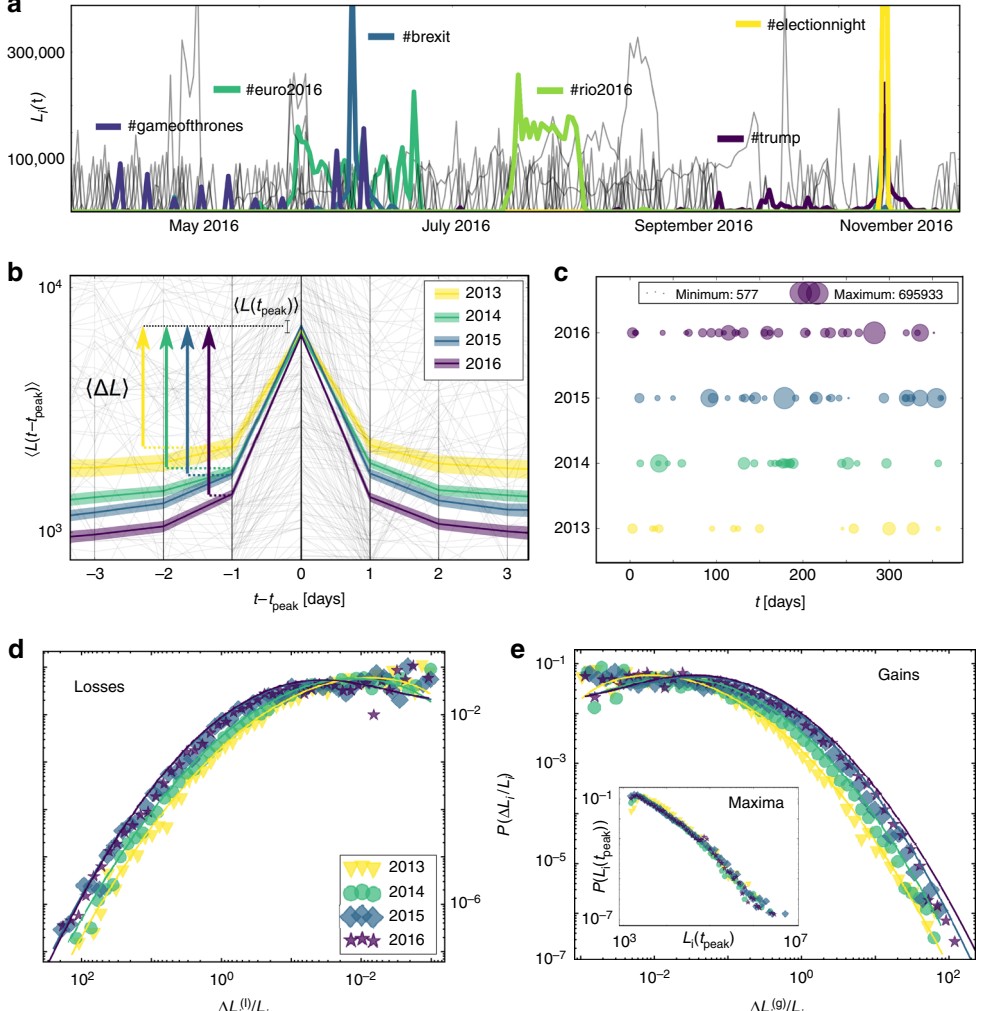

**Fig. 1** Measuring the speed of hashtag dynamics: **a** Daily usage $L_i(t)$ of the top 50 hashtags $i$ on Twitter in 2016, with exemplary highlighted major events. **b** Average trajectories $\langle L(t - t_{\text{peak}}) \rangle$ leading to a local maximum $L_i(t_{\text{peak}})$ in all top hashtags from 2013 to 2016. The shaded area shows the 95% confidence intervals (CI). In the background a 1% random sample of trajectories is shown in grey. **c** Temporal density of extreme events (local maxima with a lower threshold for the slopes, for details see Methods section), scattered over 365 days. **d** Distribution of daily relative losses $P(\Delta L_i^{(l)}/L_i)$ and **e** daily relative gains $P(\Delta L_i^{(g)}/L_i)$. The solid lines show fitted log-normal distributions $P(x) = 1/(x\sigma\sqrt{2\pi})\exp\left[-(\ln(x) - \mu)^2/(2\sigma^2)\right]$ (distributions truncated at $10^{-3}$). Inset: The corresponding distributions of the local maxima of each hashtag trajectory $P(L_i(t_{\text{peak}}))$ (distributions truncated at $10^3$). Source data are provided as a Source Data file and in an online repository (see data availability statement)

**Acceleration across different domains**. The acceleration is not unique to Twitter. We observe similar developments in a number of domains, online and offline, covering multiple time scales. We measure various proxies for the collective attention given to individual cultural items in different data sources. We assume that whenever a topic is discussed (hashtags on Twitter, comments on Reddit, n-grams in books, citations of papers) or consumed (tickets for movies, queries on Google), it receives a small fraction of the available attention. Generally, we use the highest available granularity, considering yearly episodes whenever possible. In all datasets we focus on top groups, analogously to the Twitter analysis (see Supplementary Tables 1 and 2, Fig. 2 and Supplementary Figs. 3–7 for more details).

More specifically: the books data consists of the yearly occurrences of individual *n*-grams ($n = 1, …, 5$), normalized by the number of books they appeared in, from the Google books corpus[5]. The values are aggregated for periods of 20 years, to achieve good statistics despite the coarse granularity of the data. The movie data contains weekly box-office sales of Hollywood

movies in the US, normalized by the number of theaters they were screened in, from Box office Mojo[29]. We use weekly data to avoid accounting for increases in sales towards the weekends. To reach reasonable sample size we aggregate the weekly data for periods of 5 years over the last four decades. The Google data comprises weekly search queries for individual topics from the monthly top charts normalized by the maximum within each category, from Google Trends[30]. Since the data only covers eight years we aggregate over a period of one year to obtain sufficient temporal granularity of measurements. The Reddit data consists of daily comment counts in the discussions attached to individual submissions on Reddit[31], aggregated yearly from 2010 to 2015. The data regarding scientific publications is quantified by monthly citation counts of individual papers from the APS corpus[32], aggregated for periods of 5 years, due to the coarser granularity, covering the last three decades. Wikipedia is described by the daily traffic on individual Wikipedia articles[33], aggregated yearly from 2012 to 2017.

Despite the diversity of the datasets in terms of contextual background, timespan or size, Fig. 2 and Supplementary Figs. 2, 3

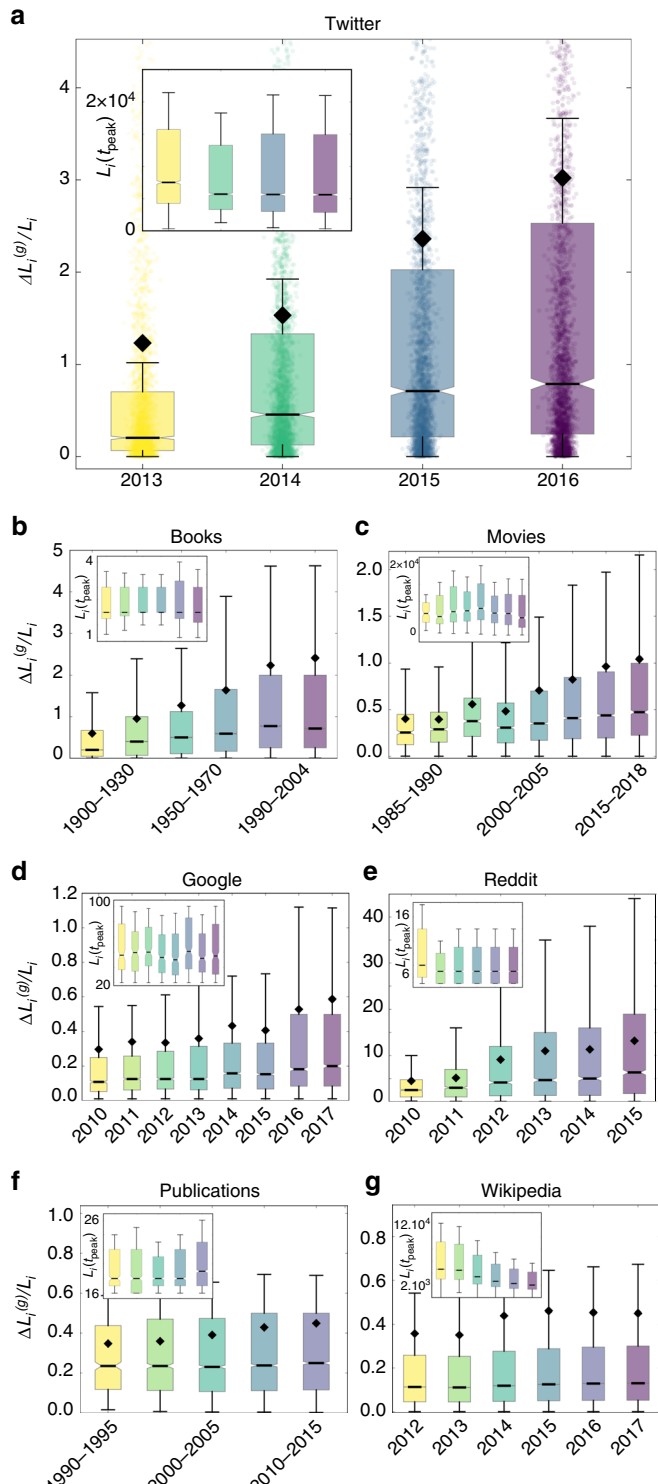

**Fig. 2** Box-plot representation of relative gains in different data sets: The values of the relative gains $\Delta L_i^{(g)}/L_i$, shown in a box-and-whisker representation for **a** Twitter, **b** Books, **c** Movies, **d** Google, **e** Reddit, **f** Publications, and **g** Wikipedia. The median is shown as a black bar and the mean as a black diamond. Whiskers are chosen to show the 1.5 of the interquartile range. In **a** the data points are included for illustration, in the other plots they are not shown. The insets show the same representation of the peak values $L_i(t_{peak})$ over the same time intervals (without mean value). Source data are provided as a Source Data file

and 4 show how the relative changes $\Delta L/L$ are similarly distributed, suggesting common underlying mechanisms. As noted above, the relative character of the observable $\Delta L/L$ compensates for many structural differences such as system size or normalization factors, enabling us to compare these systematic changes across different domains.

The box-plots in Fig. 2 show how the respective distributions of relative gains $P(\Delta L_i^{(g)}/L_i)$ undergo a long-term development in almost all datasets (Supplementary Figs. 3 and 4 for full distributions and Supplementary Tables 4–10 for the fitted parameter of the log-normal distributions). The growth of the median value and its increasing distance to the mean, as well as the growing upper boxes and whiskers describe a shift and an increasing skewness of the distributions. This represents, in agreement with observations for Twitter, a development towards overall steeper slopes and more extreme changes in collective attention for individual cultural items.

It is important to note that we do not observe this development in an all considered datasets. For scientific citations and traffic on Wikipedia pages, there is negligible growth of relative gains and losses (Fig. 2f, g). A possible explanation is that the driving mechanisms in these datasets are qualitatively different; that these knowledge-based systems are less governed by popularity or attention dynamics than the other data sources (see Supplementary note 1 for further discussion).

While we observe an increasing rate of change across systems, as it was the case for Twitter, the distributions of peak popularities $P(L_i(t_{peak}))$ remain stable. The development of the peak heights do not follow a clear trend (insets of Fig. 2) and for nearly all datasets they remain stable over time (see Supplementary Fig. 5 for full distributions, the exception of Wikipedia is discussed in Supplementary note 1). These robust observations across the data sources are further affirmed by visualizing burst events over time (Supplementary Figure 6) and decreasing average times between these events (Supplementary Figure 7).

The empirical work suggests three key findings: first, the peak heights of momentary popularity for individual topics are stable over time, second, the attention associated with individual topics rises and falls with increasing gradients and third, the shifts of collective attention between topics occur more frequent. This leads to broadening and shifting of distributions of relative changes in positive and negative direction. These observations are robust across a range of systems related to dynamics of collective attention, while knowledge communication systems appear to show much weaker developments.

**Topics compete for finite collective attention**. To interpret these results, we formulate a deliberately minimal model that still captures key features of our empirical observations, with global parameters that we can interpret qualitatively in terms of real-world systems. This bare-bones model allows us to systematically investigate how the different empirical observations are connected and how they arise from system-level changes. The model is based on Lotka–Volterra dynamics for species (topics) competing for a common resource (attention)[25,26]. It describes the growth and shrinking of content volume $L_i(t)$. The growth is degraded by the competition with existing content about the same topic $i$ from the past and by the competition with other topics $j$

$$\frac{dL_i(t)}{dt} = r_p L_i(t) \left( 1 - \frac{r_c}{K} \int_{-\infty}^{t} e^{-\alpha(t-t')} L_i(t') dt' - c \sum_{j=1, j \neq i}^{N} L_j(t) \right).$$

(1)

The dynamics of each topic $i$ are governed by three basic mechanisms, proportional growth by imitation $r_p L_i(t)$,

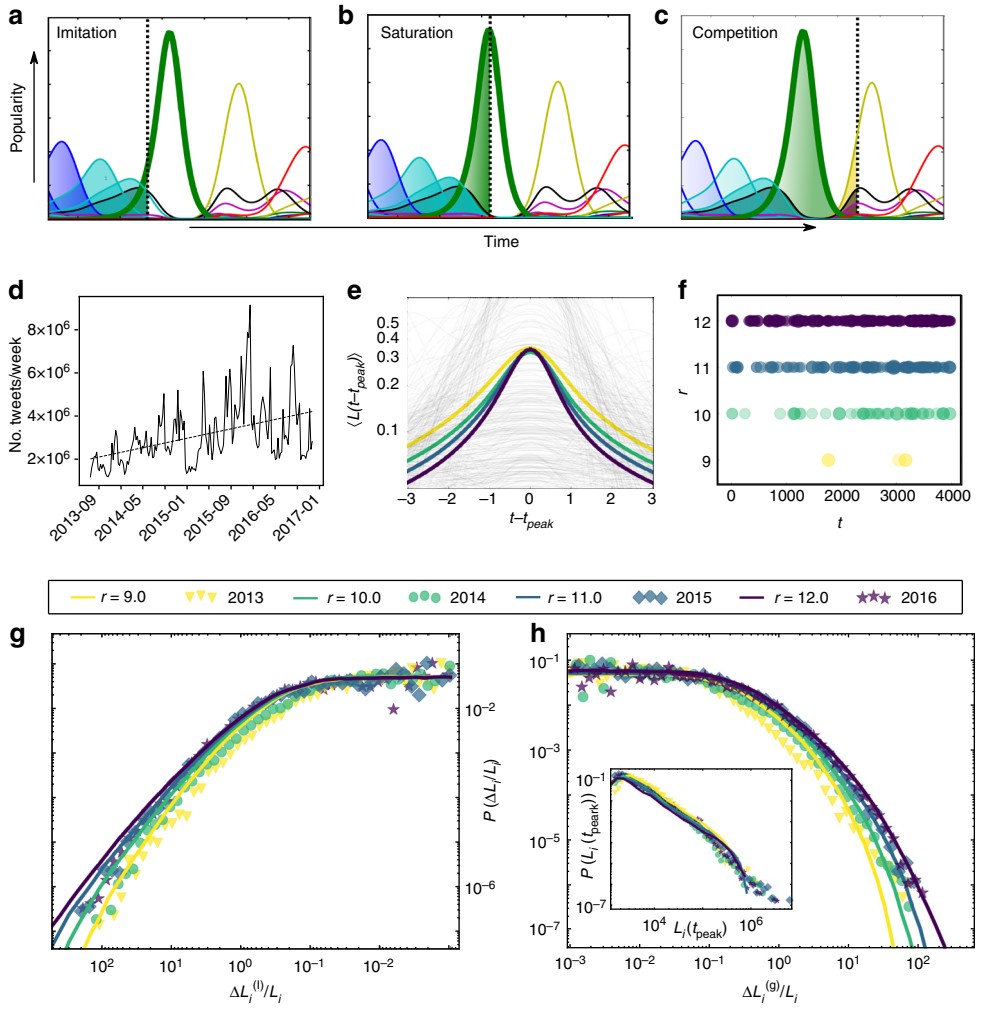

**Fig. 3** Modeling attention dynamics: **a–c** Exemplary details of trajectories resulting from Eq. (1), imitation, saturation and competition. **d** Total amount of tweets observed per week that contain hashtags among the 50 most commonly used hashtags. Their number grows from two million in late 2013 to four million by the end of 2016. **e** Average trajectories $\langle L(t - t_{\text{peak}}) \rangle$ leading to a local maximum $L_i(t_{\text{peak}})$ from the numerical solution of Eq. (1) with $\alpha = 0.005$, $c = 2.4$ and $N = 300$ and varying $r_c = r_p = r$ from 9.0 in yellow to 12.0 in dark blue, as indicated in the legend. **f** Intensity of bursts and timing. The radii of the circles encode the height of the peaks of extreme events. **g** Distribution of losses from the simulations (lines), compared with the empirical data from Twitter (symbols) and **h** distribution of gains (distributions truncated at $10^{-3}$). The inset shows the overlap of the distributions of peak heights $P(L_i(t_{\text{peak}}))$ (simulated values are rescaled and truncated at $10^3$)

self-inhibiting saturation $\frac{r_c}{K} \int_{-\infty}^{t} e^{-\alpha(t-t')} L_i(t')dt'$ and competition with other topics $c \sum_{j=1, j \neq i}^{N} L_j(t)$.

The exponential decay with rate $\alpha$ accounts for finite memory of the public, as recently observed[13]. Content regarding the topic $i$ that has been produced in the recent past is consumed and contributes to its boringness. Occurrences in the distant past do not contribute, allowing for the reoccurrence of topics. The quantity $K$ accounts for the carrying capacity from the classical population growth models and sets (together with the rates $r_p$ and $r_c$) the upper limit of growth. For simplicity, we set this parameter to 1 throughout the numerical analysis. The coupling strength $c$ accounts for the amount of overlapping context. The lower $c$ is, the more topics can coexist. For high values, the competition is fierce and topics can only grow if others decline.

The communication rates $r_p$ and $r_c$ are central in interpreting the model. Both are per capita growth rates and have the dimension of content per time. They label the rate of content creation ($r_p$, production) and the rate of consumption of the existing content ($r_c$, consumption). Considering the amount of information transfer over time is crucial because there is

strong evidence that this quantity is increasing in real-world systems.

The resulting dynamics can be separated into three phases within each of which a single mechanism is dominant: First, due to its novelty a topic gains advantage over existing topics and driven by imitation, the topic grows rapidly with rate $r_p$ (green curve in Fig. 3a). Second, at some point a large amount of content on that topic exists, which is consumed at rate $r_c$, exhausting the available attention for this topic. The boringness effect becomes dominant and attenuates its popularity (Fig. 3b). Third, as the topic's popularity decreases, resources are freed and the competition evokes newer topics $j \neq i$ to take over (Fig. 3c).

**Faster communication causes temporal fragmentation.** In Twitter, we observe a growing rate of tweets containing top hashtags per week (see Fig. 3d). This is consistent with reports of increasing communication streams, reaching back 200 years[34], covering modern telecommunication[35] up to the era of big data, where an annual growth rate of 28% of the world's communication capacity within 20 years has been reported[3].

Increasing communication rates cause qualitative changes in the behavior of even a strongly reduced version of our model. To obtain an analytic understanding of the principal behavior of the system under the variation of the rates $r_p$ and $r_c$, we consider a single topic without memory decay. Setting $c = 0$ and $\alpha \to 0$ we investigate the dynamics of an isolated topic presented to a audience with infinite memory. This single-topic case is analytically solvable and yields $L(t) = \frac{2K}{r_c} \frac{r_p e^{-r_p(t-t_{peak})}}{\left(1+e^{-r_p(t-t_{peak})}\right)^2}$ (see Methods section for details). The solution for a single topic resembles the derivative of a logistic growth process, known e.g., for the adopter dynamics in the theory of innovation diffusion[23]. This supports our interpretation of a growth process on a finite market of available collective attention. While increasing the growth rate $r_p$ is directly related to our empirical measurements of $\Delta L/L$, the development of the consumption rate $r_c$ requires a more implicit argument. The maximum of $L(t)$ for the isolated topic depends on the ratio $r_p/r_c$ (see Methods section for details). Consequently, the stable peak heights in the empirical observation suggest that the rates of content production and consumption might be different, but increasing $r_p$ requires a proportional increase of $r_c$ to maintain this property (for simplicity we choose $r_c = r_p \equiv r$). The parameter $r$ controls the slopes, positive and negative; the peak heights stay stable but are reached more rapidly, again consistent with real-world observations (see Supplementary Figure 8a).

This yields the following interpretation of the empirical findings for individual topics: The rate with which new content is created increases in proportion to the rate we consume content and exhaust attention (saturation). This causes gains of popularity to become steeper, while the saturation point is also reached more quickly. Thus, the peak height is conserved, but it is reached earlier and the phases of collective attention are shortened.

Returning to the empirical data, we consider a more realistic scenario of multiple, competing topics: The fully coupled system with finite memory effects Eq. (1) i.e., $c > 0$ and $\alpha > 0$, is solved numerically. The dynamics can now become complex, exhibit chaotic behavior[36,37], and characteristics of self-organized criticality[38,39], with broad distributions of event sizes (see Supplementary note 2 and Supplementary Figure 9).

With the multitopic model in Eq. (1), we can now fit the model's parameters to the gain distribution from 2016 and obtain an excellent fit with a Kolmogorov–Smirnov (KS) distance of 0.01 and a $p$ value of 0.85 (see Methods section and Supplementary Table 11 for details). Remarkably, changing a single parameter $r$ for the in- and out-flux of information allows us to reproduce all other observations at once, for Twitter (Fig. 3e–h) and for the other datasets by the same procedure (Supplementary Figs. 10 and 11). The agreement with real-world data and the simplicity of their interpretation make the results particularly persuasive.

All aspects of our observations on the various datasets are reproduced by the model. In Fig. 3e, the average trajectories show the same development as in the empirical findings (Fig. 1b). The shortened popularity phases cause a more rapid release of resources to the competitors and topics rise up more frequently (Fig. 3f). This corresponds to the observations in Fig. 1c and Supplementary Fig. 6 (Supplementary note 3 for an analytic approximation of the frequency). Importantly, none of the model's other parameters preserves the peak heights and simultaneously increases the slopes under variation as $r$ does. Supplementary Figure 12 shows that shorter memory (increasing $\alpha$) leads to higher peaks, stronger competition (larger $c$) lowers the overall curve. Similarly an increasing number of overall discussed topics (larger $N$) decrease slopes and peaks.

For a visual comparison of model and data, we overlay the shapes of the distributions of gains, losses and maxima in Fig. 3g, h.

The excellent fit is confirmed through quantitative measures (see Supplementary Table 11) and is robust to alternative measures, such as the logarithmic changes (see Supplementary Figure 13). Thus, by fitting a single global communication rate we are able to capture the changing distributions of gains, losses, and peak heights, simultaneously.

## Discussion

Over the past decades, many parts of modern society have become digitized, enabling researchers to study longitudinal changes to the dynamics of collective behavior. We have focused on content dynamics of cultural items, and our research supports the common experience[40] of ever faster flows of collective attention. The empirical findings we present are remarkably robust across many domains of public interest, covering a wide range of different time scales. They reveal significant changes in the statistical properties of collective attention dynamics. The gradients of content trajectories become steeper, but their peaks remain relatively stable, causing periods of individual popularity to become shorter.

Our modeling suggests that shorter attention cycles are mainly driven by increasing information flows, represented as content production and consumption rates. As influx increases, individual topics are adopted more rapidly, leading to steeper rises in collective attention with the self-inhibitory effects of saturation resulting in an equally steepening downfall. Thus, in our modeling framework, producing and consuming more content results in shortening of attention spans for individual topics and higher turnover rates between popular cultural items. In other words, the ever-present competition for recency and the abundance of information leads to the squeezing of more topics in the same time intervals as the result of limitations of the available collective attention[11].

A quantitative understanding of the factors behind the acceleration of allocation of collective attention has the potential to mitigate negative developments in modern communication systems caused by inflationary information flows. We expect these insights will spark research into the interplay between social acceleration and the fragmentation of public discourse and its potentially negative consequences[41–43].

## Methods

**Empirical analysis**. In order to measure the acceleration in our diverse ensembles of trajectories, we use statistical methods (e.g., ensemble averages) as a rough estimate, and the distributions of various observables for a more detailed picture. Generally, we analyze groups of the most popular items, because we aim to observe topics that reach societal-scale awareness and avoid small and heterogeneous subgroups. This choice allows us to use a well-mixed modeling approach, without the need to account for possibly complex network structures in the onset of lesser known topics. This choice also assures a constant number of parallel events in each measurement. We choose group sizes that are particular to each data source according to their size and to achieve reasonable data densities. We tested for possible artifacts from the choice of absolute top groups, by considering relative top groups and thresholds, and found the developments reported in the main text to be robust against these choices (see Supplementary Figure 1).

The temporally discrete nature of empirical data requires us to chose a reasonable bin size to aggregate the timestamped occurrences. The minimal size we use is 24 h, starting from 00:00 and reaching to 23:59, to avoid confusing popularity dynamics with diurnal rhythms. If the data is only available in coarser granularity, we choose that binning size.

The aggregation windows of values, used to obtain the distributions for comparison, are chosen in a similar way. We generally compare distributions year-by-year and more generally, we choose aggregation windows for the losses, gains, etc. in the following way: if the data granularity is available on a daily level, we stick to the choice of yearly aggregation (e.g., in Twitter, Google, or Reddit). When the granularity of data points is coarser, e.g., weekly or yearly, we used wider aggregation windows, such as 5 or 20 years (e.g., Movies or Publications). This choice results in sample sizes that allow us to extract smooth distributions and cover the available time frame evenly.

Some of the observables depend on market size or infrastructure and cover long time scales. Whenever possible we choose observables to be independent of these

influences. For example, we consider normalized proxies for popularity as in box-office sales and Google Books.

**Sampling techniques**. To measure collective attention we focus on popular and globally visible cultural items. On Twitter the acceleration developments appear weaker the lower the items are ranked (Supplementary Figure 1). A possible explanation for this behavior is additional mechanisms driving smaller popularity peaks, that are influenced differently, e.g., by social network structures or daily rhythms and less by the global news-spreading velocity. To focus on what we call collective attention we choose the following popularity-related samples. Twitter: From the set of hourly aggregated hashtag occurrences from altogether 43 billion individual tweets we sample the 50 most used of every hour. The resulting sample sizes are listed in extended data Supplementary Table 2. The raw sampling counts already hint at the observed development: the total number of different hashtags ranked in the hourly top 50 within a year increases from 2013 to 2016. In 2013, for example, a hashtag has been part of the set of the top ranked hashtags on average for the period of $(365 \times 24 \times 50)/25031 = 17.5$ hours, while in 2016 it stayed only $(365 \times 24 \times 50)/36703 = 11.9$ hours. Hence, we can observe a development towards higher turnover rates in the popular discussion. This observation as well as the other results (Figs. 1 and 2) are stable with respect to changes in the size of the top group and become even more pronounced within the highest ranked hashtags (e.g., in the top ten of each hour, see Supplementary Figure 1).

For books, we extract the 1000 most frequently used $n$-grams from each year ($n = 1, ..., 5$), normalized by the number of books in which they appear. A $n$-gram is a specific sequence of $n$ character strings. Especially for higher $n$ such $n$-grams are likely to correspond to a specific turn of phrase, related to a specific topic, if they are used several times per book. In this case the occurrences of phrases in books are taken to be a proxy for the attention that their authors spend on certain topics. Similarly to the usage of specific hashtags we take such phrases as representations of topics of collective attention. The normalization per book allows a measurement that corresponds to the relative volume of a phrase usage without the total growth of the book market. Within the 20 year bins used in our investigation, we observe a similar effect in the raw counts as described above for Twitter: The number of different $n$-grams within the yearly top group over 20 years grows as listed in Supplementary Table 2 (the last number is smaller due to the shorter observation window of just 14 years).

As another example from the offline world we collected box-office sales (i.e., sold ticket prices) of popular Hollywood movies from the past four decades. This measurement is another quantification of the collective attention towards cultural items in form of movies. The more people view a given movie, the more collective attention it received. We average values by the number of theaters in which each movie is screened. This avoids a bias due to a growing number of cinemas. We use 4000 individual movies from the weekly top list of Box office Mojo[29]. Similar to previous datasets, the number of individual movies within the same time interval increases (Supplementary Table 2) (the last observation window is only 3 years).

For Google, we chose the top 20 search terms from every month in the period of 2005 to 2015 as they are listed from Google Trends topcharts. We use different categories, namely: people, songs, actors, cars, brands, TV shows and sports teams. We interpret the number of web-enquirers concerning these items as collective attention on them. In each category, five queries can be compared at once and their volume is normalized to the maximum among them. On the one hand this provides a normalization, but on the other hand the limitations from Google Trends imply that these values are not perfectly comparable. We overcome this by defining relative gains and losses as independent from system size, but note that, in this case, the distribution of maximum values in Supplementary Figure 5c is still non-trivial to interpret.

On Reddit we use as a popularity proxy the volume of discussion on submissions. The more a submission is discussed the more people are actively engaged by spending their attention to the particular topic. We extract the number of comments on each submission from the corpus of a 10% random sample from all comments on Reddit and focused on the top 1000 commented submissions of each month. The growing sample sizes again confirm the increasing turnover rates.

From the body of papers in the APS corpus we analyzed the citation counts of papers with more than 15 citations within a month. Whenever researches cite another work they are discussing a related idea, and by that focusing their attention on that topic. In this dataset any acceleration of citation dynamics is visible only for highly cited papers, we speculate that this because only the mechanisms of, e.g., imitation apply in this case (detailed discussion see Supplementary notes).

We also include the 100 most visited articles of every hour on the English Wikipedia in our analysis. Again, the active visit of a Wikipedia article is an act of focusing ones attention on the topic, described by the article. In the case of Wikipedia, we do not observe a pronounced development towards larger turnover rates within this definition of top articles. This is corroborated by the lack of strong changes in the other observables for this dataset (discussion see Supplementary notes).

**Average trajectories**. From the dynamics of $L_i(t)$ of an entity $i$, we compute the ensemble average of the trajectories around local maxima $L_i(t_{peak})$ as a first step. The grey lines in Fig. 1b show 1% of of -trajectories $L_i(t)$ around a local maximum, contributing to an intuitive visualization of the dynamics in the Twitter dataset. The colored lines are the ensemble average values of all sub-trajectories $L_i(t_m \pm t)$

from each trajectory $L_i(t)$ around all its $M_i$ local maxima, at respective times $t_1 \dots t_{M_i}$ for each year: $\langle L(t_m \pm t) \rangle = \sum_{i=1}^{N} (\sum_{m=1}^{M_i} L_i(t_m \pm t)/M_i)/N$. Evaluated three timesteps before and after the maxima, we observe long-term changes in the average values $\langle \Delta L \rangle$ and a relatively stable average height $\langle L(t_{peak}) \rangle$. Following this insight we analyze and compare the statistical distributions of all the relative changes $\Delta L_i/L_i$ in each time step and the maxima $L_i(t_{peak})$ with respect to their long-term developments.

**Distribution of relative changes $P(\Delta L_i/L_i)$**. For a more refined picture on the system level than the average of $L_i(t)$ around a local maximum, we examine the changes in popularity at every given point in time. In order to avoid trivial effects such as system size dependence, we do not consider the total changes $\Delta L_i$ for measuring the velocity of attention allocation, but the relative gains $\Delta L_i/L_i$ in positive direction: $[\Delta L_i^{(g)}/L_i](t) = (L_i(t) - L_i(t-1))/L_i(t-1) > 0$ at every instance $t$. Additionally we define the losses, symmetrically by reversing the order of events: $[\Delta L_i^{(l)}/L_i](t) = (L_i(t) - L_i(t+1))/L_i(t+1) > 0$. The normalized empirical distribution of relative losses $P(\Delta L_i^{(l)}/L_i)$ is plotted in Figs. 1d and 3g with an inverted $x$-axis to visually clarify their negative character (in Supplementary Figure 4 for the other datasets without that visualization). The distributions of gains $P(\Delta L_i^{(g)}/L_i)$ are shown in Figs. 1e, 2, S3, and 3h. All plots are in double logarithmic scale and distributed among log-scaled bins.

**Fitting continuous distributions**. Finding a good fit for the distribution described above is not the central point of this work, but the correct fit contributes to a better understanding of the data. In addition, the fitted parameters reveal the visually observed developments quantitatively as listed in Supplementary Tables 4–10. To find the best suited probability density function we use maximum likelihood fitting[44] considering an array of well known continuous distribution functions, namely: exponential, power-law, log-logistic (Fisk), Cauchy, normal, gamma, Pareto, logistic, uniform, Weibull, power-law with exponential cutoff, and log-normal. Supplementary Figure 2 shows the empirical data and the fitted candidate functions. For the optimal choice of distribution we consider the KS-statistics in order to quantify the goodness-of-fit. As listed in Supplementary Table 3, the log-normal distribution is the best choice in most cases and is plotted using a thick red line. Other good candidates are the Pareto, the Weibull, and the log-logistic distribution. In the case of some of the datasets, these provide an even better fit (e.g., Pareto for Wikipedia). For simplicity we use the log-normal distribution: $P(x) = 1/(x\sigma\sqrt{2\pi}) \exp\left[-(\ln(x) - \mu)^2/(2\sigma^2)\right]$ in all cases. The corresponding KS-statistics are generally very small, they are listed alongside the detailed fitted values in Supplementary Tables 4–10.

Since it is not the main point of this work to identify the exact shape of the distributions, we use box-and-whisker plots for the representation in Fig. 2. Specifically, we use notched box plot representations, with median values shown as black bars and mean values as black diamonds. Whiskers are chosen to show the 1.5 of the interquartile range. The position of the median relative to the mean and the size of the upper box and whisker are good measures for the right-skewness of the data.

**Distribution of maximal popularity $P(L_i(t_{peak}))$**. To quantify the observation of the average peak heights, the inset of Fig. 1e and S5 show the distribution $P(L_i(t_{peak}))$ of values of local maxima $L_i(t_{peak})$ from each trajectory. These plots are in double logarithmic scale with log-scaled bins. Corresponding box plots as described above are shown in the insets of Fig. 2. Their shape has various characteristics, but generally the maxima are broadly distributed in all data.

**Statistical testing**. For the distributions of gains and losses we fitted a log-normal distribution to the data. This choice is based on the comparison of many candidate functions as described above and shown in Supplementary Figure 2. To check the goodness-of-fit we use the two-sided Kolmogorov–Smirnov (KS) test[45] for continuous functions, which return low statistics values (Supplementary Tables 4–10) for all fits. The $p$ values (for the hypothesis that the data is drawn from a log-normal distribution) vary largely and are generally small. In some datasets small sample sizes can contribute to the low-$p$ values. As mentioned above the aim here is not to find the optimal probability density function to describe our data but to get an estimate for its broadness.

We also us the two-sided KS test for two samples to compare the distributions we obtain from the simulation to the empirically observed distributions. The distribution we aim to fit with the chosen parameters ($\alpha = 0.005$, $c = 2.4$, $N = 300$, and $r = 12$) is the gain distribution from Twitter in 2016. We reach a KS-distance of 0.01 and a $p$ value of 0.85 (for the hypothesis that the two sets are drawn from the same distribution, Supplementary Table 11). Additionally we find that the distributions from the other datasets agree very well with our simulations (see Supplementary Figs. 10 and 11, KS-statistics in the caption).

**Burst events**. Another way to demonstrate the development of accelerating media is to visualize the density of extreme events in collective attention dynamics. This can also be understood as a visualization of the broadening distributions of relative changes from Supplementary Figs. 3 and 4. We define this as an relative increase

above a threshold, directly followed by a steep decline. These events can be understood as bursts or spikes. To define such an event in Twitter we detect a relative gain $[\Delta L_i^{(g)}/L_i](t_{burst})>5$ followed by a loss $[\Delta L_i^{(l)}/L_i](t_{burst})>5$. For the other datasets we adjust these values to obtain a reasonable density of events for visualization (Google Books: 12, 12; Movies:1.5, 1.5; Google Trends: 2, 2; Reddit: 25, 25; Citations: 1, 1; Wikipedia: 35, 35). These values depend on the broadness of the gain and loss distributions and are chosen to track only the most extreme events (furthest points in the tail). The resulting bursts are plotted in Supplementary Figure 6 as scatter plots over time, for more recent times from bottom to top. The resulting average heights at the burst event $\langle L(t_{burst})\rangle$ and the time interval $\langle\tau\rangle$ between them are shown in Supplementary Figure 7. The developments are robust to the choice of slope thresholds.

**Reducing dimensionality**. The proposed model consists of $N$ coupled distributed-delay differential equations that can be transformed into a system of $2N$ ordinary differential equations. Introducing the auxiliary variable $Y_i(t) \equiv \int_{-\infty}^{t} e^{-\alpha(t-t')}L_i(t')dt'$ in Eq. (1) of the main text and applying the Leibniz rule, we obtain the transformed system:

$$\frac{dL_i(t)}{dt} = r_p L_i(t)\left(1 - \frac{r_c}{K}Y_i(t) - c\sum_{j=1,j\neq i}^{N} L_j(t)\right), \quad (2)$$

$$\frac{dY_i(t)}{dt} = L_i(t) - \alpha Y_i(t). \quad (3)$$

This system can be solved numerically, using standard solvers like the Runge–Kutta method. The parameters for our simulations for the Twitter dataset are $N = 300$, $\alpha = 0.005$, $c = 2.4$, $K = 1.0$ and using the condition for stable peak heights $r_p \sim r_c$ we set $r_c = r_p \equiv r$, which we varied within a range $r \in \{9.0, 10.0, 11.0, 12.0\}$.

The interpretation of the numerical parameter values is not in the focus of the work, but they were optimized to fit the empirical distributions. The system size $N = 300$ contributes to a richer dynamic than using only, e.g., $N = 50$, the value of the memory decay $\alpha$ and the global competition coupling $c$ are difficult to measure in empirical data, but a possible investigation in future research. The growth rates of Fig. 3d can not be mapped directly to the rate $r_p$, because we measure global production rates, while $r_p$ describes a per capita growth per piece of content for each topic, but also this can be approached in the future. Generally, the qualitative results are very robust within wide ranges in the parameter space and the fitting is possible without extensively fine tuning the parameters.

After starting with random initial conditions ($L_i(0) \in [0,1]$) and empty memory ($Y_i(0) = L_i(0)$), we allow for a transient of 500 time units in our simulations before starting the measurement. The finite memory makes self-sustaining dynamics possible without the constant introduction of new topics. Introducing additional topics could be another variant of the simulation to account explicitly for exogenous driving by ever new events, but is not in the scope of this work. Solving the equations above results in $N$ trajectories that can be analyzed for the same observables as the ones we observe in the datasets. We average over 100 realizations to obtain the statistical distributions, because ergodicity is not necessarily given. In Fig. 3a–c such trajectories are shown as examples, and in Fig. 3e the average values around local maxima are plotted. This plot is much smoother than the empirical one in Fig. 1b, because we are able to refine granularity of our time-discretization for the simulation. From this data it is straight forward to extract the extreme events (Fig. 3f), the distributions of gains and losses (Fig. 3g–h) or the values of maxima (inset Fig. 3h, values rescaled to meet empirical distribution). The simulation parameters were chosen by parameter scanning to minimize the KS-distance (0.01) of the resulting relative gain distribution to the one observed in 2016 on Twitter. The other distributions then follow from that simulation and overlap well with the data. Only the parameter $r$ is varied to fit the distributions of the earlier years. For the other datasets the same procedure is applied to achieve the fits in Supplementary Figs. 10 and 11.

**Analytic solutions for a single topic**. To gain a deeper insight to the interplay of the two rates $r_p$ and $r_c$, we consider the minimal scenario of the model, an isolated topic ($c = 0$). This reduces Eq. (1) to:

$$\frac{dL(t)}{dt} = r_p L(t)\left(1 - \frac{r_c}{K}\int_{-\infty}^{t} e^{-\alpha(t-t')}L(t')dt'\right), \quad (4)$$

choosing the limiting case of infinite memory $\alpha \to 0$ leads to

$$\frac{dL(t)}{dt} = r_p L(t)\left(1 - \frac{r_c}{K}\int_{-\infty}^{t} L(t')dt'\right), \quad (5)$$

which can be written as:

$$\frac{dL(t)}{dt} = r_p L(t)\left(1 - \frac{r_c}{K}Y(t)\right) \quad (6)$$

$$\frac{dY(t)}{dt} = L(t). \quad (7)$$

The chain rule yields

$$\frac{dL(t)}{dY(t)} = r_p\left(1 - \frac{r_c}{K}Y(t)\right), \quad (8)$$

which leads to (with the necessary condition $L(Y = 0) = 0$)

$$L(t) = r_p Y(t) - \frac{r_p r_c}{2K}Y(t)^2 = \frac{dY(t)}{dt}. \quad (9)$$

This first-order equation can be solved with the substitution $v(t) = Y(t)^{-1}$ and has the solution:

$$Y(t) = \left(\frac{r_c}{2K} + Ce^{-r_p t}\right)^{-1}. \quad (10)$$

With the condition $Y(t_{peak}) = \lim_{t\to\infty}\frac{1}{2}Y(t) = K/r_c$ for logistic growth processes, where $t_{peak}$ is the time when $L(t)$ is maximal as we used for empirical data in the main text, this yields the solution:

$$Y(t) = \left(\frac{r_c}{2K} + \frac{r_c}{2K}e^{-r_p(t-t_{peak})}\right)^{-1} = \frac{2K}{r_c}\left(1 + e^{-r_p(t-t_{peak})}\right)^{-1}, \quad (11)$$

which directly leads to the solution for $L(t)$:

$$L(t) = \frac{2K}{r_c}\frac{r_p e^{-r_p(t-t_{peak})}}{\left(1 + e^{-r_p(t-t_{peak})}\right)^2}. \quad (12)$$

The observation of stable peak heights becomes clear by considering the maximum:

$$L(t)|_{\frac{dL(t)}{dt}=0} = L\left(t_{peak}\right) = 2K\frac{r_p}{r_c}, \quad (13)$$

which is independent of $r$ under the assumption of the direct proportionality of the two rates $r_p \sim r_c \equiv r$. Eq. (9) is shown in Supplementary Figure 8a under this condition. This seems to be approximately true for the case of the full system (Eqs. (2) and (3)), as the numerical results show in Fig. 3e.

In the opposite limit $\alpha \to \infty$, corresponding to an audience without any memory, Eq. (4) becomes

$$\lim_{\alpha\to\infty}\frac{dL(t)}{dt} = r_p L(t)\left(1 - \lim_{\alpha\to\infty}\frac{r_c}{K}\int_{-\infty}^{t} e^{-\alpha(t-t')}L_i(t')dt'\right) = r_p L(t), \quad (14)$$

corresponding to an exponential growth process. In this scenario, even the present state $L(t)$ is immediately forgotten and does not contribute to the intra-specific competition, causing an unhindered growth.

## Data Availability

All datasets that support the findings of this study are publicly available and can be collected, requested or directly downloaded at: Twitter: https://figshare.com/articles/top50_hashtags_json/6007322 (Figs. 1–3). Books: http://storage.googleapis.com/books/ngrams/books/datasetsv2.html (Accessed: 2017-12-10) (Fig. 2). Movies: Courtesy of https://www.boxofficemojo.com/ (Accessed: 2018-16-07) (Fig. 2). Google: https://trends.google.com/trends/ (here we crawled the trajectories of the US-top trends of each month: https://trends.google.com/trends/topcharts (Accessed: 2018-01-10)) (categories: people, songs, actors, cars, brands, tv-shows and sports teams) (Fig. 2). Reddit: https://files.pushshift.io/reddit/comments/ (Accessed: 2017-11-02) (Fig. 2). Publications: requested for scientific purpose at https://journals.aps.org/datasets (Accessed: 2018-05-07) (Fig. 2). Wikipedia: https://dumps.wikimedia.org/other/pagecounts-ez/ (Accessed: 2017-11-05) (Fig. 2). The source data underlying Figs. 1d, e and 2a–g and Supplementary Figs. 2a–g and 3a–f are provided as a Source Data file.

## Code availability

A simple implementation of a numerical integrator (using standard methods) for the model is available in python at: https://github.com/philipplorenz/integrator_boringness.

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

## Acknowledgements
PLS and PH acknowledge the support by Deutsche Forschungsgemeinschaft (DFG) in the framework of Collaborative Research Center 910. We thank A. Koher, F. Baumann, A. Spreen, L. Ramzews and the anonymous reviewers for useful comments and fruitful discussions.

## Author contributions
All authors designed the study. PLS, BMM and SL evaluated the data. PLS and PH developed the theoretical model. All authors analyzed the results and wrote the manuscript.

## Additional information

**Competing interests:** The authors declare no competing interests.

