## [Peer Review File · Nature Communications]

Reviewer #1 (Remarks to the Author):

The resubmitted version of the paper provides an interesting combination of empirical analysis and theoretical model to study attention dynamics in several media. In my opinion, main results are that the dynamics in public attention is accelerating through years and that some statistical traits can be reproduced through a simple model. First statement is partially novel although the way how it is analysed is, as far as I know, new. In this sense, the approach to media such as Twitter is novel and potentially valuable since it provides a different perspective than the other typically offered in IT and media field.

If one careful focusses on answers to the third referee, the authors have not made much effort to answer the referee's concerns. It is true that the authors provide different data sets but it is also true that the data analysis in the core paper is very much relied on Twitter data. Additionally, there is no much reflection about what is meant by attention when talking about the different data been analysed. Additionally, a pshychologist or a sociologist can get very nervous in the way how attention is discussed and, although it can provide a new perspective, on how it is linked to for example "boringness" behaviour. Along the same lines, the authors are mixing social attention and public attention terms. To me, these are two different but subtle terms that need for a definition and for some consistency in the text and in the referee's reponse #3. Moreover, the media used and the different nature of each dataset can provide different meanings to the word attention. Further discussion is truly necessary to avoid a misuse of the term or at least to avoid a deep misunderstanding under the form of a broad statatement in the title. This reflection in the discussion section is truly necessary to guarantee that the current contribution has some impact in the study of social and news (digital) media. Such an effort

needs to go further than just providing a list of references. It needs a deeper discussion in both introduction and conclusion sections.

Just for the sake of adding few other comments not been raised by referee

#3: In relation to the methodology been used, I can see a majour concern. Being a non-stationary and multiplicative random process (velocities or variations are measured as a ratio), one should extremely cautious about raising some conclusions about averages changing every year. To avoid any byas due to the spurios drift in the underlying stochastic process, one should take the logarithmic change:

$\ln(L(t)/L(t-1))$ instead of $(L(t)-L(t-1))/L(t-1)$. Authors should at least test if results are still robust when dealing with the logarithm. People from finance did already discussed this issue in 1960s and clearly opted to

use the logarithm since then (although some physicists are still wrongly using ratios). Concerning the cumulative function, there is also an important

point: it is quite frequent to consider in this kind phenomena (extreme events) the Weibull function or the (exponentially) truncated power law.

I can guess that its performance will be similar or even better than the log-normal pdf. None of them is discussed and none of them is provided.

It is also necessary to see same plots with semi-log or without any log.

The fit could then not be seen as perfect as it is shown. The methodology description is in my opinion still incomplete and hard to follow. And still provides me some doubts. Just to mention a couple: is 24h window taken under the form of moving average? In twitter, for instance, most cited hashtags seem to be events whin a beginning and an end. Is this affecting the analysis and the peak?

Reviewer #3 (Remarks to the Author):

The manuscript argues that onsets of public attention are getting shorter with time. A variation of the Volta-Volterra model is proposed - capturing the interplay between novelty, imitation, saturation (and boredom) and available (competing) alternatives in data consumption and production. A simplified version of the model is applied on an impressive range of very different datasets from Twitter hashtags to phrases (n-grams) in books and movie sales - showing surprisingly consistent results. While average peak values are stable over time, the authors observe that the gradients are steeper. The authors show that the change can be modeled by the production-consumption ratio.

First, although the idea seems intuitive and was, in fact, studied in the past to some extent - I must say that I love the proposed model and the breadth of the datasets. Considering the variety of datasets and experimentation was well executed - this paper provides a nice perspective and I expect it to facilitate further discussion in the wider CSS community, thus I want to see it published.

I do wish, however, to see some further clarifications, mostly regarding the interpretation of the model.

In the abstract the authors present two hypotheses (a) recent developments, (b) the way information is disseminated and absorbed. The authors conclude that the findings support the latter. On the one hand, the definition of "recent developments" is not clear to me. On the other hand I'm not sure that the findings support (b). The findings do show that onsets are shorter and that they can be modeled with respect to time and amount of data produced (or better - the amount of data available for consumption). The absorption part is left somewhat open (see below).

The consumption parameter $r_{\{c\}}$ is estimated/fitted to the data (am I getting it right? If i'm wrong - how is $r_{\{c\}}$ defined/measured in practice?). This is perfectly fine, however the claim is that that this parameter represents the 'boringness'. I'm not convinced that this is indeed the the case. This is an important point since the authors not only propose a nicely fitted model but use it to validate a hypothesis in relatively complex social arena. I could imagine that the number of content producers and the fact that most user tend to consume rather than produce, and that production is not distributed equally (a small number of accounts produces a large chunk of the data) may play a role here. I wish the authors make a clearer distinction between the model and the interpretation or tone down the claim.

Another issue is the proposed model vs. the model that is actually being used (c and $\alpha = 0$). The authors report that the number of topics (N) the competition (c) and memory change the height of the peaks but not the gradients, or at least not to the extent that it depends on r . It is not clear to me whether the authors opted for a simple and more manageable model (lines 170, 192-4) or that more complex models were tried but were not found very informative (lines 196-203) . The short paragraph in 186-189 adds to this confusion.

This way or another, these results are a bit surprising and I would love seeing a more substantial discussion (and experimentation) about it.

One final note - while the authors factored out the competition term, it seems that competition may creep in via the production term as some concepts have a variety of hashtags related to the concept (some of the hashtags are interchangeable, e.g. #howIMetYourMother and #himym or #debate #presdebate #debate16, etc. I may be that instead of measuring popularity of hashtags (or ngrams) one may consider measuring the popularity of *concepts* as variability within a concept may suppress dynamics of the terms under the concept.

Similarly, again, focusing on the Twitter data, the authors use the top 50 hashtags per time slot, leaving one to wonder if the set of top hashtags are representative of the general phenomena or that there is some qualitative difference between the top k hashtags and other popular hashtags.

Misc.

- The manuscript could use another editing pass as it contains some typos, e.g. 'a audience' (line 171) or terms like 'newness' (what's wrong with novelty?) or 'boringness' (just very uncommon and weird, I'd propose 'saturation')
- At the end of the discussion section the authors refer to filter bubbles and fake news. I can speculate about the connection between the manuscript and these phenomena but the association is implicit and it feels like superficial buzzword plugging.
- Page numbers (4,9,12) float in text in the manuscript submission template.

Below we respond to reviewers in detail. For an overview of edits, see cover letter.

Response to reviewer #1

Comment: *The resubmitted version of the paper provides an interesting combination of empirical analysis and theoretical model to study attention dynamics in several media. In my opinion, main results are that the dynamics in public attention is accelerating through years and that some statistical traits can be reproduced through a simple model. First statement is partially novel although the way how it is analysed is, as far as I know, new. In this sense, the approach to media such as Twitter is novel and potentially valuable since it provides a different perspective than the other typically offered in IT and media field.*

If one careful focusses on answers to the third referee, the authors have not made much effort to answer the referee's concerns. It is true that the authors provide different data sets but it is also true that the data analysis in the core paper is very much relied on Twitter data.

Answer: We thank the reviewer for raising these points.

Firstly, concerning Twitter as the key example: In the revised version, we now emphasize that while Twitter is used as a key example, we do, in fact, perform the full analysis on all data sets. In order to provide full information regarding these results, we have added Fig. S9 and Table S3. We placed most of this additional information in the supplementary information, for readability and in order to keep the length of the main text reasonable. In addition, we note that we had previously made an effort in response to the previous third reviewer by adding two additional offline datasets (Movies and Books) to the analysis. In light of comments below, we have greatly expanded the discussion in the main and in the methods section to emphasize the important role of the breadth of datasets.

Secondly, concerning the role of Twitter as social/news media: We agree with the reviewer that Twitter is not exclusively a 'social' media platform but also a 'news' medium. In the revised version of the manuscript we have clarified the scope of the paper to focus, not on 'social acceleration', but on 'acceleration of collective attention'. With a focus on collective attention, we believe that the social network/news issue becomes less of a problem. In fact, one might see it as an advantage, because Twitter could be taken to reflect the interest of some 'public' towards various topics, such as politics or pop-culture.

Comment: *Additionally, there is no much reflection about what is meant by attention when talking about the different data been analysed. Additionally, a pshychologist or a sociologist can get very nervous in the way how attention is discussed and, although it can provide a new perspective, on how it is linked to for example "boringness" behaviour. Along the same lines, the authors are mixing social attention and public attention terms. To me, these are two different but subtle terms that need for a definition and for some consistency in the text and in the referee's reponse #3. Moreover, the media used and the different nature of each dataset can provide different meanings to the word attention. Further discussion is truly necessary to avoid a misuse of the term or at least to avoid a deep misunderstanding under the form of a broad statetement in the title.*

This reflection in the discussion section is truly necessary to guarantee that the current contribution has some impact in the study of social and news (digital) media. Such an effort needs to go further than just providing a list of references. It needs a deeper discussion in both introduction and conclusion sections.

Answer: These are crucial points and we thank the reviewer for bringing them up. A large part of our efforts in this substantially revised version of the manuscript address these matters.

Upon realizing our previous lack of clarity, we decided to use the term ‘collective attention’ to describe the phenomenon at the center of our analysis. Specifically, we now use ‘collective attention’ to describe the quantity that is spent on cultural items whenever they are discussed in the public sphere. This is the underlying idea of the concept ‘attention economy’, which we discuss in the revised version of the paper in much more detail. We now define the term explicitly in the beginning of the manuscript.

We changed the title of the manuscript accordingly and now use the terminology consistently throughout the text. In our conclusion, the scope of our findings is more strongly emphasized as well.

Finally, we have a detailed discussion regarding each of the individual interpretations of the various proxies for public attention/interest in the methods section, we are hopeful that these changes will satisfy the reviewer.

Comment: *Just for the sake of adding few other comments not been raised by referee #3: In relation to the methodology been used, I can see a major concern. Being a non-stationary and multiplicative random process (velocities or variations are measured as a ratio), one should be extremely cautious about raising some conclusions about averages changing every year. To avoid any bias due to the spurious drift in the underlying stochastic process, one should take the logarithmic change: $\ln(L(t)/L(t-1))$ instead of $(L(t)-L(t-1))/L(t-1)$. Authors should at least test if results are still robust when dealing with the logarithm. People from finance did already discuss this issue in 1960s and clearly opted to use the logarithm since then (although some physicists are still wrongly using ratios).*

Figure R1: The development of hashtag dynamics on twitter, quantified via the logarithmic change $\ln(L(t)/L(t-1))$ and the distribution of the same data as well as the results from the simulation as the logarithmic change $\ln(L(t)/L(t-1))$.

Answer: This is an excellent idea. We have tested the logarithmic change as a measure for

the changes we observe in the empirical data as well as for our simulation results. We have added the figures S12a and b to the supplementary material (c.f. Fig R1).

These figures show that the development persists when we consider the logarithmic change measurement: not only the tails, but the entire distribution is shifted. The distributions of logarithmic changes from the empirical data and the simulation fit very well. (KS-statistics: 0.09, 0.08, 0.05, 0.05).

Comment: *Concerning the cumulative function, there is also an important point: it is quite frequent to consider in this kind phenomena (extreme events) the Weibull function or the (exponentially) truncated power law. I can guess that its performance will be similar or even better than the log-normal pdf. None of them is discussed and none of them is provided.*

Answer: This is a very good point. In the updated version of the manuscript, we have expanded the methods section, to describe the maximum likelihood fitting process and the comparison of the different candidate distribution functions in more detail.

In the revised analysis we now include the two suggested distributions (see Fig. S2). The first panel of Fig. R2 shows them in direct comparison to the log-normal distribution for the Twitter example.

Figure R2: Highlighted Weibull and powerlaw with exponential cutoff distributions in comparison to the lognormal distributions, fitted to the 2016 Twitter dataset.

Additionally we added a Tab. S3 that lists all KS-statistics as a quality measure for the fit for all distribution functions and datasets under consideration. Here, it becomes clear that the log-normal distribution is the overall best fitting function, thus we use this function to study the development of its parameters over time. The Fisk, the Pareto, and the Weibull distributions are also well suited candidate-distributions. We now mention this fact explicitly in the caption and highlight this in the second panel of Fig. R2.

Comment: *It is also necessary to see some plots with semi-log or without any log. The fit could then not be seen as perfect as it is shown.*

Answer: Again, we agree. We have now added Fig. S12c and d to address this issue (c.f. Fig. R3). Indeed, using this representation the lower values in the distribution can be investigated in more detail and show agreement with the simulation in this range, too. We also

would like to point to tables S3, S11 and the caption of Fig. S8 for an additional quantification of the goodness of the various fits via the KS-statistics.

Figure R3: The empirical data and the simulated distributions, as shown in Fig. 3h, but using semi-logarithmic and with linear axis.

Comment: *The methodology description is in my opinion still incomplete and hard to follow. And still provides me some doubts. Just to mention a couple: is 24h window taken under the form of moving average? In twitter, for instance, most cited hashtags seem to be events when a beginning and an end. Is this affecting the analysis and the peak?*

Answer: We agree that the methods section required a thorough revision.

The methods section has been substantially revised and expanded (c.f. the attached diff-file). A few, major issues are listed below:

Firstly, we now more thoroughly discuss the details of the empirical measurements and explicitly mention that we do not use a sliding window to cover 24 hour time spans, but that we use simple time-bins from 00:00 to 23:59. As explained in the main text, we make this choice even if finer granularity is available in the data, since we are interested in the dynamics of the public interest towards e.g. political topics and not how diurnal rhythms might impact dynamics.

Secondly, the potential issue related to cut-off effects by a finite top-group raised by the referee is very important. Therefore, we now include additional tests and a brief discussion in the revised manuscript to rule out any artifacts that could arise from the specific sampling of a top group. Specifically, we included two different relative top-group samplings in order to rule out observing a changing proportion of the system, as well as a threshold sampling (see Fig. S1 and Fig. R4 below for the reviewer's convenience).

Summarizing, we sincerely thank the reviewer for the constructive remarks. We genuinely believe that the manuscript has been improved substantially, especially in terms of the discussion, and the insightful feedback has also resulted in a streamlining of the focus and scope of the paper. Finally the new analysis strengthens the robustness of the empirical results.

Figure R4: Different alternative sampling methods and the resulting average trajectories around local maxima.

Response to reviewer #2

Comment: *The authors attempt to measure the pace of social life and show that it has been accelerating via quantitative analysis of trend duration in several data sets. The idea is interesting and important, and their analysis is interesting and much improved compared to their previous draft. I fully support publication of this manuscript in Nature Communications (I really like to see it published!) if the authors can address my remaining concerns detailed below, but note that this will require another round of significant revision.*

I number my comments below as in my response to the authors' original manuscript, but I replace my comment (and their response) with "OK" if their response and/or revisions fully addressed my concerns.

Answer: We are very thankful for the positive remarks of the reviewer and the support for publication.

We also want to mention how grateful we are for the depth of the review as well as the level of detail of the remarks, and the great suggestions. It is truly unusual to receive feedback of this caliber: There is no doubt that these comments have greatly helped to further improve our work and to lift it to a new level of clarity and quality.

In the following, we will respond to each point individually, and hope that the reviewer finds that our answers a) meet the high quality of feedback we have received, and b) address all of the reviewer's concerns appropriately.

We have also produced a version of the manuscript color-highlighting all changes (diff.pdf).

Comment: *Detailed suggestions/comments:*

3. Original lines 20-21: Does L count the total number of appearances of the hashtag, or the number of distinct users of the hashtag?

Response: The observable $L_i(t)$ refers to the number of appearance of the hashtag i in time step t . This is now stated more clearly in the main text.

I would have preferred that the authors explain this when they introduce the symbol L . It still doesn't seem like it's clearly communicated to me.

Answer: We agree that the introduction of the observable L was still unclear and we have revised that passage.

Comment: *4. Line 21: Why use the symbol " L " for hashtag counts? Presumably there is some word that the authors have in mind? Maybe "length?"*

Response: We use L in reference to 'likes', which is an established word for popularity of online content.

Why not state this when you introduce the symbol? Maybe I'm a bit slow with these things, but that word did not occur to me when I thought of hashtags—I was really puzzled.

Answer: Good point. We now explicitly mention this connection in the revised manuscript.

Comment: 11. Figure 1b: What are the gray lines? These can't be all hashtags for the 4 year period, right? And why do they appear, in some cases, to have local peaks at $t_{\text{peak}} - 2$ or $t_{\text{peak}} + 2$? Also, the x-axis needs units indicated (presumably days).

Response: While the color curves show the average for each year, the gray curves are a one percent, random sample shown for illustration. The unit of the x-axis is days, indeed, which has been added in the revised figure.

Again, the text doesn't explain that this is a 1% random sample as far as I can tell. Why not state that in the caption?

Answer: Agreed. We have added an explanation statement in the caption of Fig. 1b.

Comment: 14. Figure 1d: the x-axis is not a function of time. Notation is again misleading: this should be evaluated at $t = t_{\text{peak}} + 1$. Y-axis needs numbers and a label - it could also be labeled "abundance" or "probability" or "PDF" or " $P(\Delta L/L)$ " or something similar as the authors see fit. Why is x-axis arranged (contrary to near- universal convention) left-to-right from larger to smaller?

Response: We inverted the x-axis in Fig. 1d for illustration of the 'negative' character of the losses and to highlight the symmetry between gains and losses (by that the labels of the y-axis in Fig. 1d have been moved to the right axis and share the label with Fig. 1e). We clarified the notation and the labels in Figs. 1d and 1e, which refer to the distribution of losses and gains in each timestep. See also response to comment 9.

*I don't find the symmetry so surprising. As I understand it, your relative gains and relative losses are both approximations of the same infinitesimal quantity $(dL/dt) * (dt/L) = dL/L$, but with a minus sign difference. This is the case if you replace the discrete time step 1 in their definitions by an arbitrary time step Δt and then evaluate the limit as $\Delta t \rightarrow 0+$. Other than that (which maybe doesn't need to be addressed by the authors unless they wish to), I'm satisfied with the revised figure.*

Answer: We thank the reviewer for this remark. It is true that these observables describe the same quantity dL/L in the limit of continuous time. When we speak about gains $[\Delta L_i^{(g)}/L_i](t) = (L_i(t) - L_i(t - 1))/L_i(t - 1) > 0$ or losses $[\Delta L_i^{(l)}/L_i](t) = (L_i(t) - L_i(t + 1))/L_i(t + 1) > 0$ we always mean only values that are positive in either (time-)direction (as indicated). By that we split the quantity dL/L in two parts, which are distributed very similarly (in discrete time) in all datasets. We find this to be a non-trivial insight.

Comment: 21. Lines 34-35: Log-normal was the best fit among what set of possible functions Tested?

Response: We agree that the manuscript lacked a discussion of the systematic comparison of possible probability distribution functions. We added a section in the methods section for this discussion and compare various candidate functions for all datasets in Fig. S2. The log-normal function showed to be the overall best candidate function, but we emphasise that

finding the optimal fit is not in the focus of the work, but rather provides a quantification of the changes in the distributions, which are quantified by the fitted parameters in Tabs. S3-S9 This is much improved.

Two comments: (1) On line 324 of the revised manuscript, you say "maximal" where I believe you meant "minimal." (2) To compare fits to distributions with varying numbers of parameters, an approach should generally be used that penalizes models with more free parameters, e.g., by using the Aikake Information Criterion (AIC). But this is clearly not the focus of the paper, and I don't think there's any need to delve further into the comparison of fits.

Answer: We thank the reviewer for the positive evaluation of the improvement and generally agree that it is ideal to use principled methods to determine the best fit (also correcting for the number of free parameters) whenever possible.

However, as the reviewer points out, optimal fits are not the focus of the paper. In order to provide additional quantitative information and at least to be able to use the KS-tests consistently throughout the paper for a goodness-of-fit evaluation, we added Tab. S3 that list all candidate distribution functions and their test results. In this sense the best-fit distributions still provides insight to the character of the data and the long-term developments (as Tabs. S4-S10 show). Furthermore, in light of comments from reviewer #1, we have also added the Weibull distribution and the power-law distribution with an exponential cutoff function to the comparison.

Comment: *30. Line 82 / equation 1: This equation has some confusing dimensional issues. The lhs has units [hashtags/time], but the rhs is not clear to me. Presumably r has units of [1/time] so that the quantity outside the large parentheses should fully account for units and the parenthetical expression should be dimensionless. But how can the second term in that expression (the integral term) be dimensionless? The exponential can't have units, and thus the integral should have units [hashtags*time]. Something seems wrong: maybe a dimensional prefactor needs to be inserted. (When only one hashtag is present, the equation has a fixed point at $L = \alpha$, which again emphasizes the unit problem.)*

*Response: We thank the reviewer for pointing out this important issue. It is true that the integral term in Eq. (1) has units [hashtags*time]. We have now introduced the second rate r_c to correct that. Interestingly the newly introduced rate adds interpretation to the model, because it can be understood as the rate at which existing content from the recent past is consumed and exhaust the available attention for the topic. We found (analytically) the condition for the observed stability of peak heights to be a constant ratio of the two rates r_o and r_c . Incorporating this condition we redid all the simulations with an adjusted competition parameter c and could achieve good results.*

This model revision is an improvement, but I'm unhappy to say that there are still problems with dimensional consistency. Not all terms on the RHS have the same units ([hashtags/time]), so I think the model needs to be revised again, despite the significant work you guys obviously put in after the last round of review. This next revision could be fairly minimal (replacing r_c with r_c/L_{infty} , though that would imply some necessary modification of your description and analysis) or a bit more significant, but I can't support publication of the paper with this problem still present. I'm attaching a supplement to this review demonstrating the problem and suggesting a solution.

Answer: We are truly grateful for this important discussion by the reviewer.

We fully agree that the dimensionality issues of the RHS of the model had persisted. In the revised manuscript we have now implemented many suggestions made in the supplementary material to the review. In particular, we now include a carrying capacity K in all equations, which addresses the dimensionality problem. We also added a description and adjusted the analytic solution for ($\alpha \rightarrow 0$), as well as the height of the maxima.

We also changed the lower integration bound to $-\infty$ in our analysis, as suggested by the reviewer. We included the corrections regarding the missing integration constant in Eq. (9) (we renamed r_o to r_p for ‘production’-rate:

$$Y(t) = \left(\frac{r_c}{2K} + C e^{-r_p t} \right)^{-1}. \quad (1)$$

With the condition $Y(t_{\text{peak}}) = \lim_{t \rightarrow \infty} \frac{1}{2} Y(t) = K/r_c$ for logistic growth processes, where t_{peak} is the time when $L(t)$ is maximal as we used for empirical data in the main text, this yields the solution:

$$Y(t) = \left(\frac{r_c}{2K} + \frac{r_c}{2K} e^{-r_p(t-t_{\text{peak}})} \right)^{-1} = \frac{2K}{r_c} \left(1 + e^{-r_p(t-t_{\text{peak}})} \right)^{-1}, \quad (2)$$

which directly leads to the solution for $L(t)$:

$$L(t) = \frac{2K}{r_c} \frac{r_p e^{-r_p(t-t_{\text{peak}})}}{(1 + e^{-r_p(t-t_{\text{peak}})})^2} \quad (3)$$

and very elegantly, as suggested by the reviewer directly yields the height of the maximum:

$$L(t) \Big|_{\frac{dL(t)}{dt}=0} = L(t_{\text{peak}}) = 2K \frac{r_p}{r_c}. \quad (4)$$

This remark led elegantly to the correction of our solution and the unit problem.

Furthermore, we consider the limit $\alpha \rightarrow \infty$, which (in our formulation) yields exponential growth, since the present value $L(t)$ is also forgotten immediately. For the numerical evaluation of the full model, we set the carrying capacity to $K = 1$.

We are very thankful for the reviewer’s suggestion to use the normalization condition of the decay kernel to obtain a rate in front of the distributed delay part of the model a natural way. In fact, we redid all the analysis and simulations including this approach. Ultimately, however, we realized this approach implies a mechanism that we do not aim to implement in our framework and the numerical results suggest that there are aspects of the empirical observations that are not reproduced with the approach suggested by the reviewer. In the following, we will elaborate on these aspects in more detail.

The main concern is that by introducing the normalization, the memory decay and the consumption rate are no longer independent. A main point of the paper is that we can reproduce the observations by a relatively simple increase of only one parameter which determines the overall content production and consumption. The suggested modification means that not **more** content is consumed when increasing the rate. Rather, it implies that content is consumed **earlier**, which is not our intention (see Fig. R5 for an illustration of the decay kernels).

Additionally We wanted to maintain a stable shape of the decay, because we believe it is governed by human factors that might change on longer timescales (c.f. Ref. [13]).

Figure R5: The two different decay kernels under the variation of the consumption rate.

In order to ensure that we had considered the reviewer’s elegant suggestions fully, we reran all numerical simulations and optimized the parameters in the case of the Twitter data. The parameters to fit the 2016 dataset we found are: $\alpha = 0.008$, $c = 0.004$, $r_c = 6.80$, $L_{inf} = 1.26$. We decreased r_p and α in the same way as in the paper (with $r_p/\alpha = \text{const.}$) and obtained a good fit for the gain and loss distribution (see Fig. R6).

Figure R6: Gain and loss distributions simulated via the suggested model, compared to the Twitter dataset.

The problem becomes clear, however, only when checking on the distribution of maxima as we do in the paper. This is important because we found relative stable peak heights in the empirical data. The condition for their stability is nicely derived by the reviewer in his/her supplementary material, but it requires a Taylor expansion.

When considering at the numerical results there is a point where the approximation might break down, because the frequency of very large peak heights increases with r (c.f. first panel of Fig. R7). This is in contradiction to our observation and does not fit the results from our original formulation of the model.

Another difference becomes clear when examining the average trajectory around local maxima. Increasing alpha leads to higher frequencies and by that causes the upwards development on the right side (see second panel of Fig. R7).

Figure R7: The distribution of the heights of local maxima from the suggested model, compared to the Twitter dataset. The average trajectories around local maxima from the suggested model.

Thus, even though we like the elegant suggestion for a normalized decay kernel proposed by the reviewer a lot, it implies a dependency that we do not aim to model and which does not reproduce the observations in the minimalistic way that the original model does. For future work, we will keep this approach certainly present.

Comment: 31. Lines 101-102: Are the graphs in extended Fig 3 just showing integration of equation (1), or something else? What's the "new model" referred to in its caption? How are new topics sparked: are new hashtags introduced in the simulation, or is the total number fixed (so what's shown is just a result of the initial condition)?

Response: We agree that the description of this figure was largely incomplete and have added a detailed description in the in the supplementary notes. The first row of the figure (now Fig. S10) shows results of the classical Lotka-Volterra equations for competitive species, with a special set of parameters, which causes chaotic behavior found in reference [31]. The second row shows solutions of the same equation and parameters but with the boringness term added, which we introduced in our modeling. The chaotic behavior is stable over larger ranges in parameter space.

The new figure S10 has an error in the equation reference in its caption. Also references to it earlier in the SI incorrectly reference Fig. S8.

Answer: Thank you. We have corrected the incorrect references.

Comment: 32. Line 112 and subsequent discussion: I would like more justification for why r should be a proxy for the speed of information input/output. Clearly it has the right units, but the model imposes that hashtag changes happen on that time scale, so it doesn't seem so surprising that changing r changes the time scale of the Dynamics.

Response: Following the comment number 30 we have added an explicit consumption (input) rate r_c in Eq. (1), which provides a clearer interpretation of the model. The rate r_o determines the velocity the growth only in the beginning of a hashtags uprise, but after a short time the influence of the distributed delay hinders the growth and finally the downfall begins. We can solve the model for a minimalistic case, which clarifies the role of the rates in the dynamics

and shows that they cannot be easily be absorbed in a rescaled unit time, due to the distributed delay term.

This needs to be addressed again in light of the new problem with dimensional homogeneity mentioned above.

Answer: Since we keep the formulation of two distinct rates r_c and α , our earlier response stays largely valid. The rate r_p does not just determine the overall timescale. Because of the delay term it cannot be absorbed into the time. We understand r_p as the growth rate of the content production process that is fueled by the current popularity $L(t)$ and r_c the rate at which the existing content from now and the past consumes the available attention.

Comment: 37. *Figure 4f: Agreement with the model does not seem very good here.*

Response: We agree that the agreement is not perfect in this plot, which is due to the very specific shape of a two phase power-law of the empirical data, while the model produces a constant power law over several orders of magnitude. When comparing the distributions of local maxima in the revised Fig. 3e, the data is distributed more regularly and the agreement becomes better.

The comparison now shown in Fig. 3e is different—it's very hard to see how theory compares to data there. Your previous fig. 4f is now the inset in panel h, and the agreement with the model appears better than before, though it's very hard to tell (I had to zoom in on the pdf with a factor of 600% to really see it). Even if agreement is not great, I don't see that as a major problem: the model is quite simple, and it still appears to capture much of what's happening in the data set. I wonder if there's another place to put this panel though—maybe separately in methods or in the supplementary information? (also: the value of r_c is not reported in the caption).

Answer: We have added a figure in the supplementary material (Fig. S13) that contains some details regarding the model and a magnification of the inset of Fig. 3h.

Comment: *Some new issues in the revised manuscript: (don't worry, they're minor)*

General: The manuscript now appears to provide all the information necessary for another researcher to reproduce the reported results. Ideally, the authors would provide their final data sets (and perhaps code) to make that reproduction (and further exploration) easier. This isn't a requirement for publication, but I hope that at some point soon it will become one.

Answer: We added a source file that contains all raw data underlying the box-plots for all datasets and a small simulation software for solving the model numerically.

Comment: 60. *Line 172: Not surprisingly, your analytical solution is dimensionally inconsistent. (r_c can't be added to something without units)*

Answer: We agree and we have corrected the inconsistencies.

Comment: 61. *Fig. 3 panel e inset: Why aren't the peaks aligned at a single time?*

Answer: This was an error following from the unit inconsistency in Eq. (9) that we corrected in the revised manuscript. We have moved the inset to Fig. S13 with the corrected and aligned peaks, since it does not contribute significantly to the message of the main paper and – in light of the comments of reviewer #3 – did not contribute to the clarity of the manuscript.

Comment: 62. *Fig. 3 caption: no value is given for r_c . The same dimensionally inconsistent formula from line 172 is given. The symbol ω is used but not defined anywhere in the main text.*

Answer: We agree. We now mention explicitly that $r_c = r_p = r$ in the caption now and removed the inconsistent equation along with the unused symbol ω .

Comment: 63. *Line 411: There should be a constant of integration present in equation (9)—this somehow disappeared. It can be recast in terms of the initial condition of Y at $t=0$, or in terms of the peak time. This is the reason for item 60 above. See my supplemental review notes.*

See also pdf about dimensions of equations.

Answer: We corrected for the inconsistencies, see also Answer to remark 30.

Response to reviewer #3

We thank the reviewer for this constructive and positive review. Below, a point-by-point response to the points raised by the reviewer. We have also produced a version of the manuscript color-highlighting all changes (diff.pdf).

Comment: *The manuscript argues that onsets of public attention are getting shorter with time. A variation of the Volta-Volterra model is proposed - capturing the interplay between novelty, imitation, saturation (and boredom) and available (competing) alternatives in data consumption and production. A simplified version of the model is applied on an impressive range of very different datasets from Twitter hashtags to phrases (n-grams) in books and movie sales - showing surprisingly consistent results. While average peak values are stable over time, the authors observe that the gradients are steeper. The authors show that the change can be modeled by the production-consumption ratio.*

First, although the idea seems intuitive and was, in fact, studied in the past to some extent - I must say that I love the proposed model and the breadth of the datasets. Considering the variety of datasets and experimentation was well executed - this paper provides a nice perspective and I expect it to facilitate further discussion in the wider CSS community, thus I want to see it published.

I do wish, however, to see some further clarifications, mostly regarding the interpretation of the model.

Answer: We are grateful to the reviewer for these positive remarks and overall recommendation. We hope that the changes outlined below successfully addresses the reviewer's remaining concerns. They have helped us to further improve the quality of the paper.

Comment: *In the abstract the authors present two hypotheses (a) recent developments, (b) the way information is disseminated and absorbed. The authors conclude that the findings support the latter. On the one hand, the definition of recent developments is not clear to me. On the other hand I'm not sure that the findings support (b). The findings do show that onsets are shorter and that they can be modeled with respect to time and amount of data produced (or better - the amount of data available for consumption). The absorption part is left somewhat open (see below).*

Answer: We agree that our wording was unclear in the previous version of the abstract.

In the revised version we have reformulated the abstract substantially to clarify our statement, which stated informally is: The consistent findings across the different datasets and reaching back almost 100 years can be explained by a minimal model and the variation of a single parameter for information transmission without relying on extraordinary developments. Since the progress of information transfer is a core characteristic of modernization, the model suggests that this is an 'inevitable' development.

Comment: *The consumption parameter r_c is estimated/fitted to the data (am I getting it right? If im wrong - how is r_c defined/measured in practice?). This is perfectly fine, however*

the claim is that that this parameter represents the ‘boringness’. I’m not convinced that this is indeed the the case. This is an important point since the authors not only propose a nicely fitted model but use it to validate a hypothesis in relatively complex social arena. I could imagine that the number of content producers and the fact that most user tend to consume rather than produce, and that production is not distributed equally (a small number of accounts produces a large chunk of the data) may play a role here. I wish the authors make a clearer distinction between the model and the interpretation or tone down the claim.

Answer: Thank you for making this point. The structure of our argument is the following:

The parameter r_c describes the rate at which the existing content from the current time and from the past is consumed. In the context of the model, this contributes to the ‘boringness’ of the topic, by exhausting the available attention, resulting in a negative feedback for its growth.

The second observation we make, is the stability of peak heights. This can (in the modelling framework) be explained by a constant ratio of r_p and r_c ($r_p/r_c = \text{const.}$).

From this observation we conclude the proportionality of both rates, which still allows for very different rates (e.g. $r_c \gg r_p$), because only their ratio determines the development of the peak heights. Intuitively this can be formulated as ‘the saturation of a topic happens quicker with increased production and consumption of topic, freeing available resources for new topics earlier’.

To better express this reasoning in the paper, we firstly clarified the language surrounding the models in the revised manuscript. So that we now emphasize that our model is a ‘minimal’ (the simplest possible) explanation of the observations and that its main purpose is to relate the most important mechanisms and parameters (which describe public attention dynamics) to each other.

Secondly, we performed a number changes to clarify the description of the model, see diff.pdf and the next answer.

Comment: *Another issue is the proposed model vs. the model that is actually being used (c and $\alpha = 0$). The authors report that the number of topics (N) the competition (c) and memory change the height of the peaks but not the gradients, or at least not to the extent that it depends on r . It is not clear to me whether the authors opted for a simple and more manageable model (lines 170, 192-4) or that more complex models were tried but were not found very informative (lines 196-203) . The short paragraph in 186-189 adds to this confusion.*

This way or another, these results are a bit surprising and I would love seeing a more substantial discussion (and experimentation) about it.

Answer: We thank the reviewer for pointing out this potential source of misunderstanding. The related section was indeed unclear in the previous version of the manuscript.

We have restructured that part in the revised version to emphasize that after formulating the model, we first consider the case of an ‘isolated topic’ ($c = 0$ and $\alpha = 0$) in order to gain a qualitative insight to the relation of the parameters. That limit allows for an analytical treatment of the differential equation. It provides the condition for a stable peak height in

this limit ($r_p/r_c = \text{const.}$), but not much more.

Then, for all the quantitative comparison with our observations, we use the full model as stated in Eq. (1) with $\alpha > 0$, $c > 0$ and $N > 1$. This includes Fig. 3a, b, c, e (main panel), f, g and h, as well as Fig. S8, S9 and S10. Here the condition for stable peak heights, that we obtained from the isolated topic scenario holds approximately (Fig. 3e, main panel). This latter comparison is crucial for the conclusions we draw from our simulation and thus, this clarification greatly improved the manuscript.

Comment: *One final note - while the authors factored out the competition term, it seems that competition may creep in via the production term as some concepts have a variety of hashtags related to the concept (some of the hashtags are interchangeable, e.g. #howIMetYourMother and #himym or #debate #presdebate #debate16, etc. I may be that instead of measuring popularity of hashtags (or ngrams) one may consider measuring the popularity of concepts as variability within a concept may suppress dynamics of the terms under the concept.*

Answer: We agree that this is a phenomenon in online social media that we do not account for explicitly in this paper.

On one hand we aim to minimize this type of redundancy in meaning of the hashtags by considering only the most popular ones, on the other hand we measure the relative changes of individual topics, which is independent of the system size and should therefore be robust against merging such ambiguous hashtags. This implies that stronger fragmentation of and redundancy between hashtags would not account for the observed phenomenon of steeper slopes and shorter attention spans.

A more detailed analysis of the dynamics of statistically inferred ‘topics’ is definitely interesting and the long term developments of ‘topic’ behaviors are a promising venue for further research. However, since this proper identification of topics falls into the field of natural language processing and topic modeling, we hope that the reviewer will agree that such an effort is beyond the scope of this work.

Comment: *Similarly, again, focusing on the Twitter data, the authors use the top 50 hashtags per time slot, leaving one to wonder if the set of top hashtags are representative of the general phenomena or that there is some qualitative difference between the top k hashtags and other popular hashtags.*

Answer: This point about cut-off effects by a finite top-group is very important and we now analyze this important issue in the revised manuscript. We now rule out any artifacts that could arise from the specific sampling of a top group. Specifically, we included two different relative top-group samplings in order to rule out observing a changing proportion of the system, as well as a threshold sampling (see Fig. S1 and Fig. R8 for the reviewer’s convenience).

Comment: *Misc.*

- *The manuscript could use another editing pass as it contains some typos, e.g. a audience (line*

Figure R8: Different alternative sampling methods and the resulting average trajectories around local maxima.

171) or terms like newness (whats wrong with novelty?) or boringness (just very uncommon and weird, Id propose saturation)

Answer: We have done our utmost to proofread the revised manuscript thoroughly and believe to have corrected all major issues.

Comment: - At the end of the discussion section the authors refer to filter bubbles and fake news. I can speculate about the connection between the manuscript and these phenomena but the association is implicit and it feels like superficial buzzword plugging.

Answer: Good point! We certainly do not want to give the impression of buzzword plugging, but are genuinely convinced that these phenomena are related. In the revised version, we therefore toned down the claim in order to avoid this impression and to clarify our arguments.

Comment: - Page numbers (4,9,12) float in text in the manuscript submission template.

Answer: We corrected this by decreasing the spacing of the figure captions.

REVIEWERS' COMMENTS:

Reviewer #1 (Remarks to the Author):

Just a final comment that can be considered or not since I already consider that paper deserves publication. I encourage the authors to compare and discuss differences among logarithmic (Fig S12a) and the ratios (as performed in the main paper). I would recommend to add a brief sentence explaining differences (if any or if there are'nt).

Reviewer #3 (Remarks to the Author):

I'm satisfied with the revisions and would love seeing this manuscript published!

March 1, 2019

Response to reviewers:

Reviewer #1 (Remarks to the Author):

Just a final comment that can be considered or not since I already consider that paper deserves publication. I encourage the authors to compare and discuss differences among logarithmic (Fig S12a) and the ratios (as performed in the main paper). I would recommend to add a brief sentence explaining differences (if any or if there aren't).

Response: *We thank the reviewer for the support for publication. We have added the suggested, brief discussion on the different measures as well as a comment on the robustness of our findings across both of them.*

Reviewer #3 (Remarks to the Author):

I'm satisfied with the revisions and would love seeing this manuscript published!

Response: *We thank the reviewer for the helpful remarks throughout the process and for the positive recommendation for publication.*